# Automatic Extraction of Bare Soil Land from High-Resolution Remote Sensing Images Based on Semantic Segmentation with Deep Learning

Chen He [1,2], Yalan Liu [1,2,†], Dacheng Wang [1], Shufu Liu [1], Linjun Yu [1] and Yuhuan Ren [1,*]

1    Aerospace Information Research Institute, Chinese Academy of Sciences, Beijing 100101, China
2    University of Chinese Academy of Sciences, Beijing 100049, China
*    Correspondence: renyh@aircas.ac.cn
†    These authors contributed equally to this work.

**Abstract:** Accurate monitoring of bare soil land (BSL) is an urgent need for environmental governance and optimal utilization of land resources. High-resolution imagery contains rich semantic information, which is beneficial for the recognition of objects on the ground. Simultaneously, it is susceptible to the impact of its background. We propose a semantic segmentation model, Deeplabv3+-M-CBAM, for extracting BSL. First, we replaced the Xception of Deeplabv3+ with MobileNetV2 as the backbone network to reduce the number of parameters. Second, to distinguish BSL from the background, we employed the convolutional block attention module (CBAM) via a combination of channel attention and spatial attention. For model training, we built a BSL dataset based on BJ-2 satellite images. The test result for the F1 of the model was 88.42%. Compared with Deeplabv3+, the classification accuracy improved by 8.52%, and the segmentation speed was 2.34 times faster. In addition, compared with the visual interpretation, the extraction speed improved by 11.5 times. In order to verify the transferable performance of the model, Jilin-1GXA images were used for the transfer test, and the extraction accuracies for F1, IoU, recall and precision were 86.07%, 87.88%, 87.00% and 95.80%, respectively. All of these experiments show that Deeplabv3+-M-CBAM achieved efficient and accurate extraction results and a well transferable performance for BSL. The methodology proposed in this study exhibits its application value for the refinement of environmental governance and the surveillance of land use.

**Keywords:** bare soil land; high-resolution remote sensing imagery; semantic segmentation; deep learning; Deeplabv3+; CBAM

## 1. Introduction

Rapid growth of the population and economy, acceleration of urbanization, and unreasonable land use lead to a serious waste of land resources and ecological and environmental impacts in China. Bare soil land (BSL) is a significant source of air pollution. Large-area BSL leads to low land utilization rates, as well as ecological and environmental problems, such as dust pollution and soil erosion [1]. In recent years, the accurate monitoring of BSL is an urgent need for the refined management of urban environments and the improvement of land resources utilization.

Compared with traditional methods, remote sensing technology has obvious advantages in large-scale and dynamic automatic monitoring for BSL. The commonly used classification systems for land use/land cover (LULC) were proposed by the United States Geological Survey (USGS) [2], the Food and Agriculture Organization of the United Nations (FAO) [3], and the Chinese Academy of Sciences (CAS) [4]. For these systems, there are some differences in the number and definition of the classes. BSL is not defined in the USGS's and the CAS's classification systems. In the FAO's land cover classification system (LCCS), "bare areas" is defined, which includes bare soil and loose/shifting sands. For

current research on the classification of LULC, BSL is mostly overlooked, and the granularity of the classification involving bare land is usually coarse. According to the Chinese National Standard for Classification of Land Use Status (GB/T 2010–2017) [5], "BSL" is defined as "soil covered land in surface layer, basically without vegetation cover", and it is classified as "other land categories", including bare rock and gravel land and sandy land.

In terms of the classification and mapping of BSL based on remote sensing technology, there are mainly two types of studies. The first is based on traditional methods. For example, Tateishi [6] and Friedl [7] used supervised classification based on MODIS data to produce land cover products. The land cover type containing bare land is mostly defined as mixing exposed rock, saline–alkali land, sand and other lands. For these products, the renewal frequency, spatial resolution and the class granularity cannot meet the requirements for BSL monitoring. The second uses the bare soil index (BSI). Xu [1] constructed a BSI using 30 m resolution data from Landsat TM5. Nguyen [8] proposed a modified BSI using 15 m resolution data from Landsat 8. The calculation of the BSI depends on the shortwave infrared and midinfrared bands. With the development of remote sensors with high spatial resolution, sensors increasingly retain only four bands, including red, green, blue, and near-infrared, as well as a panchromatic band [9]. As a result, it is difficult to establish the BSI. In addition, BSL is mostly shapeless and has different sizes and broken boundaries. Therefore, it is still a challenge to extract BSL from other land cover classes using high-resolution remote sensing images.

Deep learning methods have been widely used in recent years. Compared with the classical machine learning method based on supervised classification, they have the advantages of a strong ability to extract adaptive features, high computational and reasoning speed, high transferability and end-to-end learning. Therefore, they are more suitable for the classification of large volumes of high-resolution images. There are many studies on semantic segmentation for the extraction of buildings [10], roads [11], water bodies [12], etc., using deep learning. In addition, Karra [13] applied a deep learning method using 10 m resolution Sentinel-2 data to produce a global LULC map. However, semantic segmentation requires a significant amount of annotated data, which limits the use of deep learning models. To address this challenge, transfer learning is proposed. Transfer learning can be divided into instance-based transfer learning, feature-based transfer learning, model-based transfer learning and relation-based transfer learning [14]. Domain adaptation is another term commonly used in transfer learning, and many studies address this challenge of limited annotated data [15–18]. For cases where the dataset types of the source and target domains are homogeneous (for example, photos of roads in different countries), domain adaptation can transfer their domain invariant features. If the data types of the source and target domains are heterogeneous (for example, photos taken by a phone and remote sensing images), model-based transfer learning is more feasible.

As a typical deep learning model for semantic segmentation, DeepLab was developed in four versions, V1, V2, V3 and V3+, from 2015 to 2018. For DeepLab V1 [19], convolution in full convolutional networks (FCNs) is replaced with atrous convolution to expand the receptive field. For DeepLab V2 [20], atrous spatial pyramid pooling (ASPP) is introduced. It allows for the input image at arbitrary scales to be performed by feature maps. For DeepLab V3 [21], several atrous convolution modules with different expansion rates are used to capture the multiscale context. Simultaneously, the model removes the full connection condition random field (CRF), which has a lesser effect on the model. Deeplabv3+ [22] was released in 2018, which uses ASPP to fuse the multiscale information in its encoder, and its concise decoder can efficiently recover the precision edge. There are some successful applications in semantic segmentation for high-resolution remote sensing images. Lin [23] integrated the attention mechanism module squeeze-and-excitation (SE) for channels into DeepLab V3 to alleviate the multiscale problems due to the different length–width ratios of roads so that the weights could be applied to different channels. The intersection over union (IoU) of the two classifications of road/nonroad was 84.62%. To solve the imbalanced distribution problem of samples, Ren [24] combined the Dice loss functions and the

binary cross entropy (BCE) loss functions with Deeplabv3+ so that the mean intersection over union (mIoU) for desert, road, water and other categories reached 87.74%. However, Deeplabv3+ has not been used for BSL extraction so far.

To reduce the impact of BSL on the environment, common governance measures include covering with dust-proof nets, hardening with cement and planting trees or grass. The areas where these measures are taken mainly have no dust pollution risk, so they were not treated as BSL in this study. They were regarded as the background objects. However, even if these areas have been treated, there will probably be some bare soil exposed. We named the BSL areas by the governance measure of planting trees, for short, BSL-PT. For the areas where bare soil is often, again, exposed due to the seasonal withering of grass, we named them, for short, BSL-PG. As a temporary treatment measure, coverings with dust-proof nets easily lead to the repeated exposure of BSL. Meanwhile, the spectral characteristics of BSL on high-resolution images are similar to those of buildings [25]. The distinction between them has still been a difficulty for the research of urban impervious surface extraction. Therefore, together with the BSL-PT, BSL-PG, dust-proof nets and buildings, they form a complex background that affects the extraction of BSL.

In this study, we adopted the definition of BSL in the Chinese National Standard for Classification of Land Use Status. In order to reduce the impacts on BSL extraction caused by complex backgrounds, buildings, BSL-PT and BSL-PG were regarded as the background in the process of the BSL dataset construction based on high-resolution remote sensing images. Then we constructed the Deeplabv3+–MobileNetV2–convolutional block attention module (Deeplabv3+-M-CBAM) based on Deeplabv3+ for the real-time semantic segmentation of BSL. Finally, it was tested on two different sources of data to verify its transferability.

## 2. Materials and Methods

### 2.1. Study Areas and Data

#### 2.1.1. Study Areas

In recent years, due to the large volume of urban renewal construction, the problem of dust pollution caused by BSL in Beijing has become prominent. Daxing District is the national collective commercial construction land pilot of Beijing. Large-scale demolition in this district has caused a large amount of BSL, which is an important source of $PM_{10}$ and $PM_{2.5}$ contributing to environmental pollution. Since 2020, Beijing has implemented the "Beijing Municipal Pollution Prevention and Control Battle Action Plan in 2020", and Daxing District has carried out monitoring of BSL using remote sensing technology.

We took three towns in Daxing District as the study areas: Yufa, Beizangcun and Weishanzhuang, as shown in Figure 1, labeled the training area, testing area and transfer area, respectively, with areas of 134.14, 47.4 and 81.33 km$^2$.

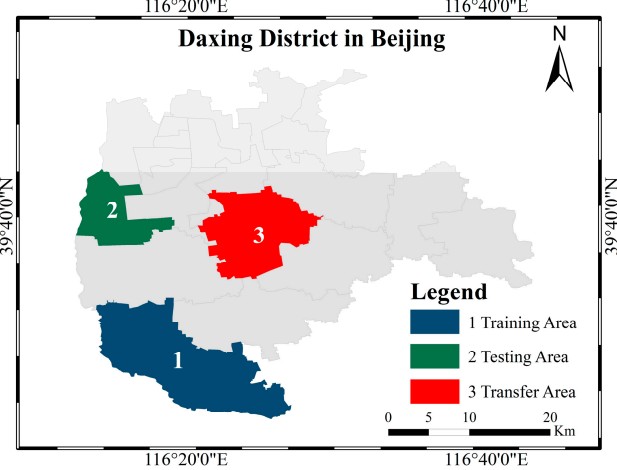

**Figure 1.** The study areas in Daxing. The coordinate system is GCS_WGS_1984.

2.1.2. Data

(1) Data sources

For this study, the images used for the above study areas all had a spatial resolution of 0.8 m without cloud coverage. For the training area and testing area, the images were acquired by the BJ-2 satellite on 21 April 2020, by Twenty First Century Aerospace Technology Co., Ltd. (Beijing, China). For the transfer area, to validate the model's transferability, the images were acquired by the Jilin-1 GXA satellite on 30 August 2021, by Chang Guang Satellite Technology Co., Ltd. (Changchun, China).

(2) Preparation for the BSL dataset

Considering that some grass or trees are planted in the BSL-governed areas, there were still some small BSL patches exposed, which should be taken as non-BSL patches. The examples are as shown in Figure 2. Some BSL-PG areas often become areas of withered grass in the winter, the textural features of which in images are smooth and mostly similar to those of BSL, such as in Figure 2a–c. Figure 2d is a field survey photo, and BSL-PT areas with dotted textures are shown in Figure 2e–g along with BSL-covered areas. Figure 2h is another field survey photo. We took BSL-PG and BSL-PT areas such as these as the background.

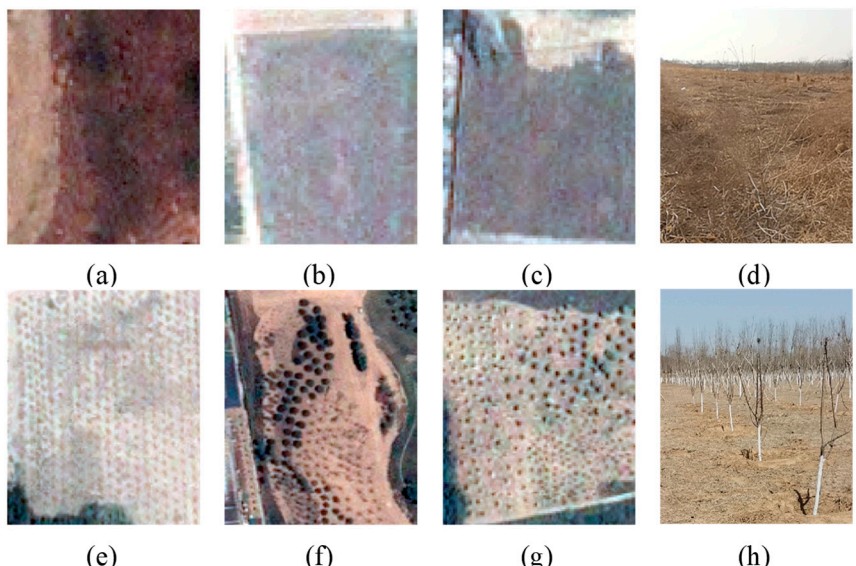

**Figure 2.** Examples of BLS-PG and BLS-PT areas from different places: (**a**–**c**) BSL-PG areas in remote sensing images; (**d**) BSL-PG areas in field survey photo; (**e**–**g**) BSL-PT areas in remote sensing images; (**h**) BSL-PT areas in field survey photo.

Generally, a deep learning dataset is divided into three parts: training set, validation set and test set. The training set and validation set are used for training the deep learning model, while the test set is used for evaluating the transferability of the model. To extract BSL, we established a BSL dataset. The procedures mainly included the preprocessing of the high-resolution satellite images, such as geometric correction and radiometric correction, image cropping and image labeling.

After cropping, the size of each image was 256 × 256 pixels, with bands of red, green and blue. Image labeling refers to the pixel-level labeling of the targets and background in the selected images. The semantic annotation was conducted in EISeg (Efficient Interactive Segmentation), which is an efficient and intelligent interactive segmentation annotation software developed based on PaddlePaddle. The BSL and its backgrounds are represented by different colors, as shown in Figure 3. The format of the images was JPG and for the labeling images PNG.

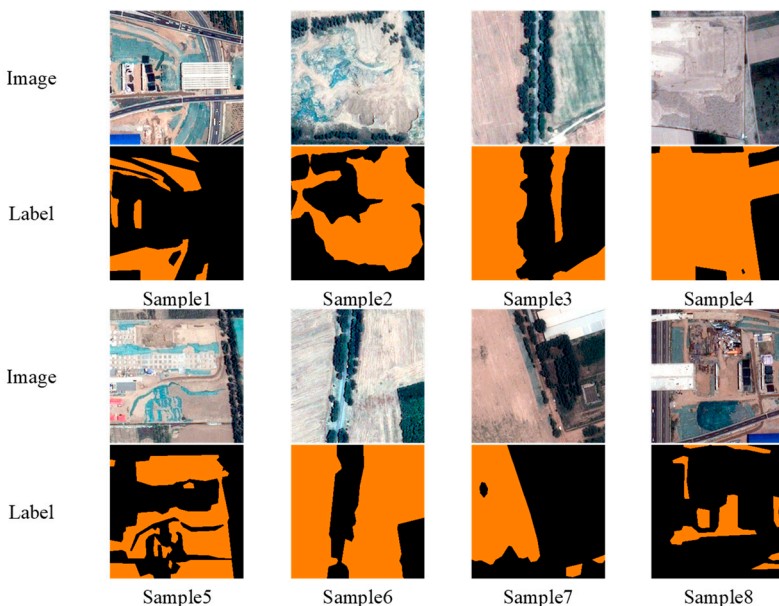

**Figure 3.** Examples of labeling BSL samples for the BSL dataset.

The final number of valid samples for the study areas was 1395, which is a small dataset with fewer than 1500 samples. Of these, 1031 samples were from training area, with 800 used for the training set and 231 for the validation set, and 364 were from the testing area and used in the test set. They are shown in Table 1.

**Table 1.** BSL dataset.

|  | **Training Set** | **Validation Set** | **Test Set** | **All** |
|---|---|---|---|---|
| Sample Number | 800 | 231 | 364 | 1395 |
| Area | Training Area | Training Area | Testing Area | - |

*2.2. Methods*

To extract the BSL from high-resolution remote sensing images, we used Deeplabv3+ as the basic model. To decrease the number of operations and the memory needed by Deeplabv3+, we replaced its Xception backbone network with a more lightweight network, MobileNetV2. To speed up the learning process of the model, we used a pretrained model on the PASCAL Visual Object Classes 2012 (VOC2012) dataset [26]. In addition, to enhance the ability of the MobileNetV2 network, the convolutional block attention module (CBAM) was used to reduce the impact of complex backgrounds. In summary, this study merged MobileNetV2 and CBAM to construct the BSL extraction model Deeplabv3+-M-CBAM. To verify the robustness and transferability of this model, a test using BJ-2 images of the testing area and a transfer test using Jilin-1GXA images for the transfer area were undertaken. The automatic thematic mapping for the two large-scale t areas was completed. The process is shown in Figure 4.

2.2.1. Deeplabv3+ Model

Deeplabv3+ is the latest semantic segmentation model of the DeepLab series, and its framework is shown in Figure 5. Deeplabv3+ employs the entire DeepLab V3 network as the encoder and uses ASPP modules and depthwise separable convolution (DSC) to fuse multiscale features and balance the accuracy and time consumption. The decoder is used to gradually recover a feature's boundary [22], which is simple but effective. In the encoder, the result of the image processing by backbone networks is divided into two parts. One is introduced to the decoder directly as the shallow feature, and the other passes through the parallel ASPP module at different scales of atrous convolution, obtaining feature extraction

results with different scales, which are merged into deep features by the compression of the features using a $1 \times 1$ convolution layer. Then, the deep features are upsampled into the decoder. In the decoder, the shallow features and the deep features can be concatenated into a merged feature map, and subsequently, the merged feature map is processed by the convolution layer and the upsampling layer. Finally, the final prediction results can be obtained.

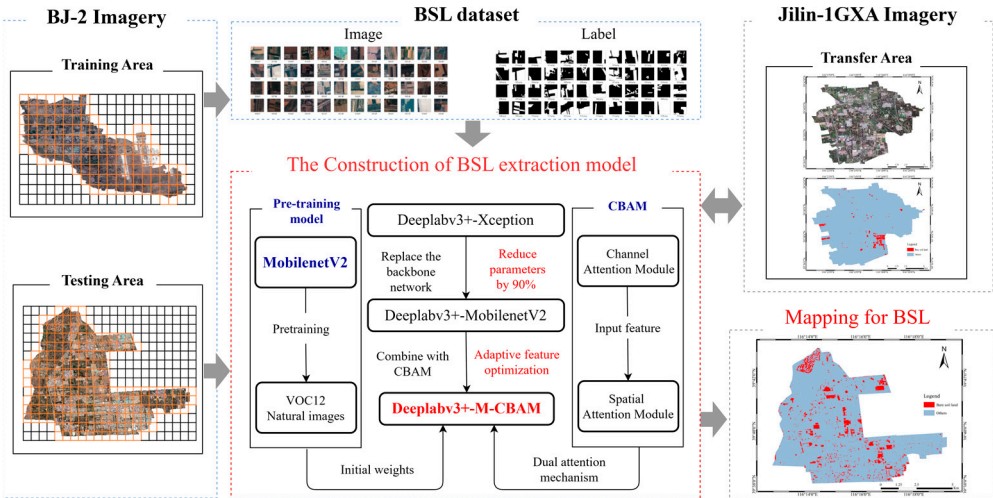

**Figure 4.** BSL extraction model framework.

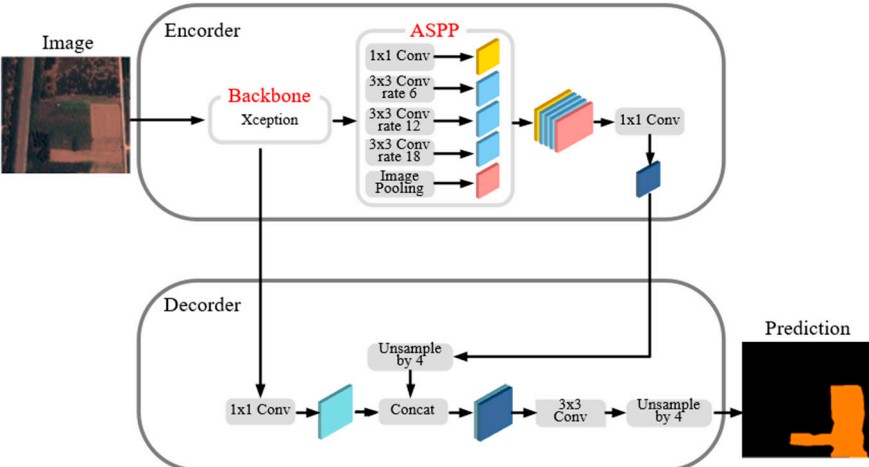

**Figure 5.** The framework of Deeplabv3+ model.

### 2.2.2. Construction of the Deeplabv3+-M-CBAM Model

Due to the BSL dataset being a small dataset, when it was used to train the Deeplabv3+ model directly for BSL extraction, it easily resulted in overfitting, which makes it difficult to achieve high robustness. Therefore, we replaced the Xception backbone network of Deeplabv3+ with the lighter MobileNetV2 to avoid overfitting by reducing the number of original model parameters. Because the background of BSL is complicated, as well as the similar spectral characteristics of buildings and BSL, we optimized the MobileNetV2 network by combining the channel and spatial attention mechanisms using the CBAM module in order to enhance the ability to distinguish BSL from BSL-PG, BSL-PT and buildings. With the above two improvements, we finally constructed the semantic segmentation model, Deeplabv3+-M-CBAM, for the extraction of BSL.

(1) A lighter backbone network—MobileNetV2

MobileNetV2 [27] is a lightweight network with a small size and strong feature extraction ability, which performs well in semantic segmentation and target detection tasks.

Its structure is shown in Figure 6. The parameters of MobileNetV2 are shown in Table 2, where "t" is the expansion factor of the input channel. Each line describes a sequence of 1 or more identical (modulo stride) layers, repeated n times. All layers in the same sequence have the same number, c, of output channels. The first layer of each sequence has a stride s, and all others use stride 1.

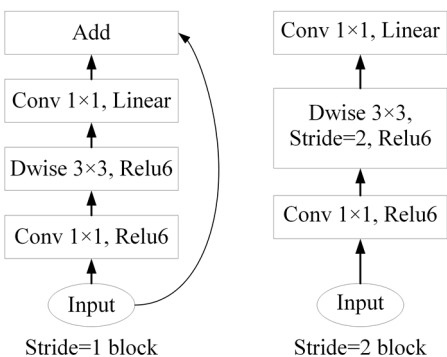

**Figure 6.** Structure of the MobileNetV2 network.

**Table 2.** Parameters of the MobileNetV2 network.

| Input | Operator | t | c | n | s |
|-------|----------|---|---|---|---|
| $224^2 \times 3$ | conv2d | - | 32 | 1 | 2 |
| $112^2 \times 32$ | bottleneck | 1 | 16 | 1 | 1 |
| $112^2 \times 16$ | bottleneck | 6 | 24 | 2 | 2 |
| $56^2 \times 24$ | bottleneck | 6 | 32 | 3 | 2 |
| $28^2 \times 32$ | bottleneck | 6 | 64 | 4 | 2 |
| $14^2 \times 64$ | bottleneck | 6 | 96 | 3 | 1 |
| $14^2 \times 96$ | bottleneck | 6 | 160 | 3 | 2 |
| $7^2 \times 160$ | bottleneck | 6 | 320 | 1 | 1 |
| $7^2 \times 320$ | conv2d $1 \times 1$ | - | 1280 | 1 | 1 |
| $7^2 \times 1280$ | avgpool $7 \times 7$ | - | - | 1 | - |
| $1 \times 1 \times 1280$ | conv2d $1 \times 1$ | - | k | - | |

When the stride for the convolution operation is 1, its input features are first processed by $1 \times 1$ convolution, which is building new features through computing linear combinations of the input channels. Next, they are extracted by the DSC layer and then processed by a $1 \times 1$ convolution layer to reduce the channels. By these processes, the results of the dimension reduction and the input feature are added to construct an inverted residual structure to alleviate the gradient disappearance. When the stride is 2, there is no inverted residual structure, because the size of the input feature is inconsistent with that of the output feature. Instead, the original information is directly merged and spliced with the subsequent results, and the other steps are consistent with the steps when the stride is 1.

(2) An optimized backbone network—M-CBAM

Since BSL-PT and BSL-PG for the governance of BSL by planting trees or grass mostly contain an amount of BSL, when using Deeplabv3+ for pixel-level semantic segmentation, this will result in the extraction of small patches of BSL in these areas, which are uninterested areas in this study. Meanwhile, buildings are difficult exclude due to the similar spectral characteristics with BSL. Hence, in order to exclude those uninterested areas, we introduced the convolutional block attention module (CBAM) [28], which can link the channel attention module and spatial attention module to improve Deeplabv3+. Its structure is shown in Figure 7.

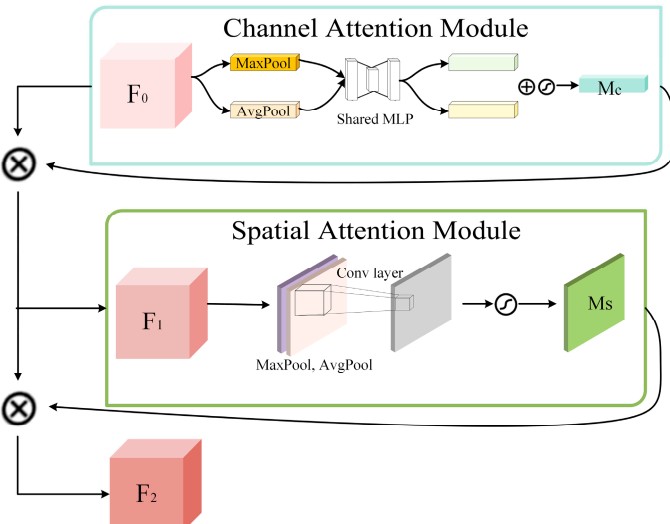

**Figure 7.** The structure of CBAM.

The channel attention module can compress the spatial dimensions of a feature map and enlarge the difference in the spectral features between different objects, solving the problem caused by the similar spectral features of buildings to those of BSL. Moreover, the spatial attention module can integrate multiscale spatial information to expand the distinction between BSL and BSL-PT and BSL-PG. Given an intermediate feature map for input, the CBAM can deduce the attentional map along two independent dimensions and then multiply the attentional map by the feature map for adaptive feature optimization. The overall attention process can be summarized as:

$$F_1 = M_c(F_0) \otimes F_0 \tag{1}$$

$$F_2 = M_s(F_1) \otimes F_1 \tag{2}$$

where $\otimes$ represents element-wise multiplication, $M_C$ is the channel attention map, $M_s$ is the spatial attention map, $F_0$ is the input feature map, and $F_1$ is the intermediate feature map.

The channel attention module compresses the feature map in the spatial dimensions to obtain a 1D channel attention map. Given $F_0$ as the input feature map, the module utilizes both max-pooling outputs and average-pooling outputs with a shared network so that the spatial information of $F_0$ is aggregated. Then, the channel attention map, $M_c$, is obtained by summing the corresponding pixels in each feature map pixel by pixel. After the element-wise multiplication of $M_c$ and $F_0$, $F_1$ is obtained. Channel attention can be expressed as:

$$
\begin{aligned}
M_c(F) &= \sigma(MLP(AvgPool(F)) + MLP(MaxPool(F))) \\
&= \sigma\left(W_1\left(W_0\left(F_{avg}^c\right)\right) + W_1(W_0(F_{max}^c))\right)
\end{aligned}
\tag{3}
$$

The spatial attention module takes $F_1$ as the input feature. After two pooling operations, the average-pooled features and max-pooled features are obtained. They are concatenated and convolved by a standard convolution layer, producing an $M_s$ of 2D. After the element-wise multiplication of $M_s$ and $F_1$, $F_2$ as the output of the CBAM is obtained. Spatial attention can be expressed as:

$$
\begin{aligned}
M_s(F) &= \sigma\left(f^{7\times7}([AvgPool(F); MaxPool(F)])\right) \\
&= \sigma\left(f^{7\times7}\left(\left[F_{avg}^s; F_{max}^s\right]\right)\right)
\end{aligned}
\tag{4}
$$

where $\sigma$ represents the sigmoid function, and $f^{7\times7}$ represents the size of the convolution kernel.

Semantic segmentation focuses on both category information ("what") and boundary information ("where"). In addition, the channel and spatial attention modules of the CBAM can learn "what" and "where" to attend in the channel and spatial axes, respectively.

In this study, we merged the CBAM and MobileNetV2 to construct MobileNetV2–CBAM (M-CBAM). The network structure is shown in Figure 8. Taking the M-CBAM as the backbone network of Deeplabv3+, the constraints on the spatial features and channel features of BSL can be added in the generation stage of the intermediate feature map. When the stride is 1, the input image will be first transformed into a feature map through the CBAM module, and then it can successively go through the 1 × 1 convolution layer, DSC layer and 1 × 1 convolution layer. The feature map of MobileNetV2 and CBAM can be fused by shortcut structure. Finally, the fusion map can be operated by the CBAM module again, and the output feature map of the backbone network can be obtained. When the stride is 2, there is no shortcut structure, and the other steps are the same as when the step size is 1, where the parameters t, n, c and s remain unchanged. In the M-CBAM structure, the category information and boundary information of BSL are enhanced twice by the CBAM.

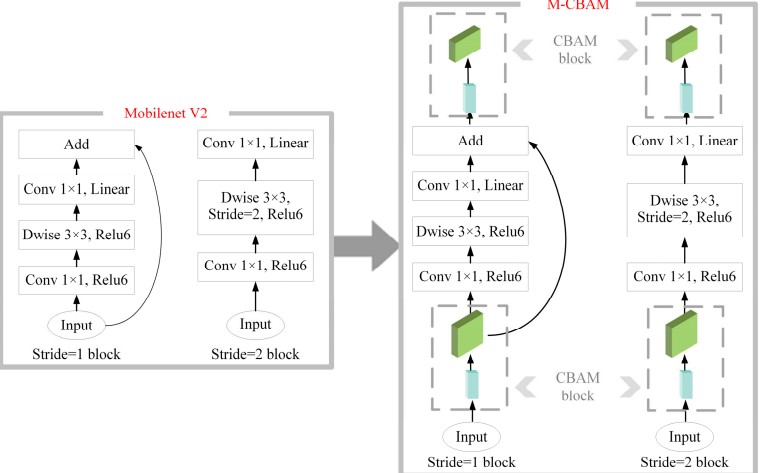

**Figure 8.** The optimization of the MobileNetV2 network.

In summary, MobileNetV2 can decrease the number of operations and memory needed by losing a small amount of precision. The CBAM can focus on BSL and suppress background information. Based on the above points, the Deeplabv3+-M-CBAM was constructed.

### 2.2.3. Model-Based Transfer Learning

Transfer learning is a method that aims to transfer knowledge from a source domain to improve a model's performance or minimize the number of labeled examples required in a target domain [14]. Model-based transfer learning is a kind of transfer learning solution. It is a way to continue learning based on the previous learning for the model. For deep learning, this refers to the fact that a model is first trained on an unrelated dataset of task A and uses the training result as the pretrained model for task B to initialize the model.

During the process of model training, if the initial weights of the model are completely random, it will take a long time for the model to find the appropriate weights and result in an insignificant effect on feature extraction, hardly achieving good network training. In order to speed up the model training, the transfer learning strategy of the pretraining model of MobileNetV2 was adopted in this study. The VOC2012 is a classical dataset for image segmentation, which contains 20 real-world categories, such as bikes, boats and people, and it is commonly used as a pretraining dataset for segmentation tasks. An effective deep learning model requires a large amount of annotated data. However, compared to large datasets, such as VOC12, the BSL dataset has a small number of labeled samples. Thus, based on the transferability of the convolutional neural network in knowledge learning [29,30], the weights learned by MobileNetV2 on VOC2012 were taken as the initial weights of the model to avoid the initial weights being too random. The process of model-based transfer learning is shown in Figure 9.

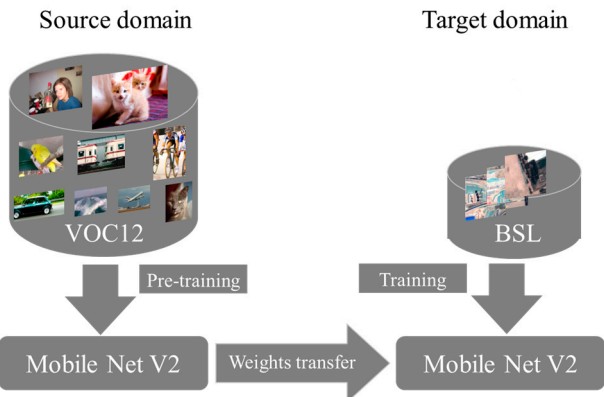

**Figure 9.** Model-based transfer learning.

2.2.4. Accuracy Evaluation Indexes of the Model

In this study, the accuracy evaluation indexes we used included *Precision*, *Recall*, *F1* and *IoU*. The evaluation index for the model's running speed was frames per second (FPS). The indexes are defined as follows:

$$Precision = \frac{TP}{TP + FP} \tag{5}$$

$$Recall = \frac{TP}{TP + FN} \tag{6}$$

$$F1 = 2 * \frac{Precision * Recall}{Precision + Recall} \tag{7}$$

$$IoU = \frac{TP}{TP + FP + FN} \tag{8}$$

In the binary classification of BSL, a BSL pixel in the label is called positive, and the background pixel in the label is called negative. In addition, a correct prediction pixel by the model is denoted as true, and wrong is denoted as false.

*TP*: true positive, where the area predicted by the model is BSL, and the truth is BSL.

*FP*: false positive, where the area predicted by the model is BSL, but the truth is the background.

*FN*: false negative, where the area predicted by the model is the background, but the truth is BSL.

*FPS* refers to the number of images processed by the model per second. Under the same software and hardware conditions, the larger the FPS, the faster the data processing speed of the model.

*2.3. Experimental Settings*

The hardware configuration of the computer used for the model training and test was an Intel Xeon e5-2678 processor, 32 GB memory and a NVIDIA Geforce RTX 2080 Ti graphics card (11 GB). The operating system was Windows10 and the main deep learning library was torch1.8.0. The programming language was Python.

In order to show the advantage of Deeplabv3+-M-CBAM, this study completed two sets of model training for Deeplabv3+-Xception (Deeplabv3+), Deeplabv3+-MobileNetV2 (Deeplabv3+-M), Deeplabv3+-M-CBAM and the classical semantic segmentation models FPN [31], UNet [32] and LinkNet [33]. The training parameters of each model are shown in Table 3.

**Table 3.** Training parameters of the different models. "-M" means using MobileNetV2 as the backbone network.

| Model | Initial Learning Rate | Optimizer | Epoch | Batch |
|---|---|---|---|---|
| FPN-M | $1 \times 10^{-4}$ | Adam | 100 | 2 |
| UNet-M | $1 \times 10^{-4}$ | Adam | 100 | 2 |
| LinkNet-M | $1 \times 10^{-4}$ | Adam | 100 | 2 |
| Deeplabv3+ | $5 \times 10^{-5}$ | Adam | 300 | 4 |
| Deeplabv3+-M | $5 \times 10^{-4}$ | Adam | 200 | 4 |
| Deeplabv3+-M-CBAM | $5 \times 10^{-4}$ | Adam | 200 | 4 |

## 3. Results

### 3.1. Model Training and Results

The learning rate decline curve of Deeplabv3+-M-CBAM is shown in Figure 10. It can be seen that the curve decline rate became increasingly smaller and finally flattened out. In addition, the learning rate dropped and was close to zero. The loss curve is shown in Figure 11. In the first round of training, the loss value of the training set differed greatly from that of the validation set, indicating that the model had not completed learning. In the 70th round of training, the two loss curves dropped to the same level and remained stable, indicating that the model training was completed. If the model was overfitted, then as the training loss curve becomes increasingly lower, the validation loss curve will become increasingly higher. The overfitting will result in poor model transferability, i.e., the model cannot be used outside of the training set area.

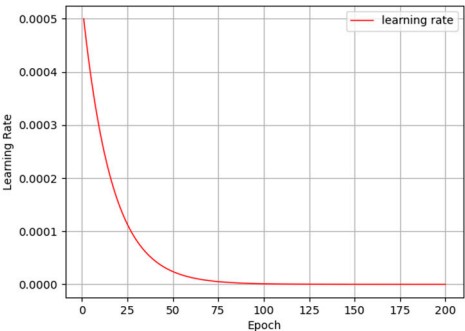

**Figure 10.** Learning rate decline curve.

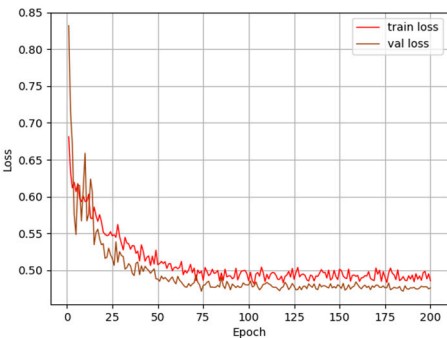

**Figure 11.** Loss decline curve.

When Xception, MobileNetV2 and M-CBAM were taken as the backbone network, their accuracy curves on the BSL validation set are shown in Figure 12a. After this training, the F1 of the Deeplabv3+ model reached 92.76%. In addition, after the first round of learning, the Deeplabv3+-M model achieved an accuracy of more than 70%, which demonstrates the advantage of lightweight networks on small datasets. After using the CBAM module to

improve the MobileNetV2 network, Deeplabv3+-M-CBAM had a higher accuracy in the early training stage, and the final training accuracy was higher than the first two models. This indicates that the CBAM module can help a model to learn the features of BSL faster and better. Compared to the combinations of MobileNet v2 with the UNet, FPN and LinkNet models, the combination of MobileNet v2 with Deeplabv3+ worked better. Their accuracy curves are shown in Figure 12a. In Figure 12b, the trends of the IoU curves and the F1 curves were similar.

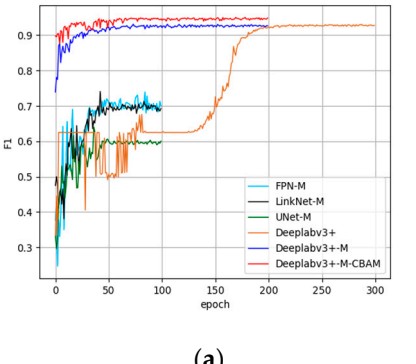
        (**a**)
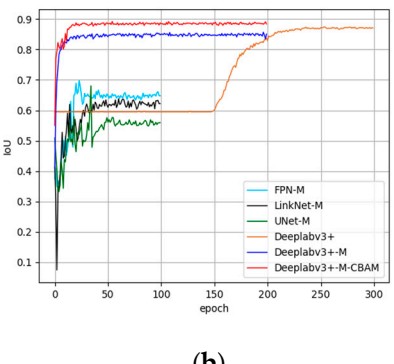
        (**b**)

**Figure 12.** Accuracy curves of the 6 models: (**a**) F1 curves on the validation set; (**b**) IoU curves on the validation set.

To make the comparison statistically meaningful, this study repeated the experiments five times and report both the average performance and the standard deviation (σ). Comparisons of the training performances of the different models on the BSL validation set are shown in Table 4. Compared with Deeplabv3+, the parameter sizes of FPN-M, UNet-M and LinkNet-M were smaller, but their accuracy was not satisfactory. When we replaced the backbone network of Deeplabv3+ with MobileNetV2, the parameter size of the model was reduced from 52.25 M to 5.60 M, which decreased greatly by approximately 90%. When combining the CBAM with the above pretraining strategy into Deeplabv3+-M-CBAM on this basis, its F1 accuracy was improved by 2.30%, from 91.09% to 93.49%, without increasing its number of parameters. Comparing Deeplabv3+-M-CBAM with Deeplabv3+-M, the FPS increased less, but compared with Deeplabv3+, the FPS increased by 2.34 times, from 17.29 to 40.50 f/s.

**Table 4.** The performance comparison for training the different models on BSL validation set.

| Indexes | / | FPN-M | UNet-M | LinkNet-M | Deeplabv3+ | Deeplabv3+-M | Deeplabv3+-M-CBAM |
|---|---|---|---|---|---|---|---|
| F1 (%) | Average | 69.44 | 61.95 | 68.32 | 91.96 | 91.09 | **93.49** |
| | σ | ±3.35 | ±2.27 | ±1.26 | ±0.76 | ±0.97 | ±0.24 |
| Precision (%) | Average | 70.71 | 63.39 | 69.67 | 92.53 | 92.49 | **93.89** |
| | σ | ±3.58 | ±1.91 | ±1.33 | ±0.48 | ±0.34 | ±0.40 |
| Recall (%) | Average | 93.58 | 92.97 | 93.17 | 92.13 | 91.64 | **94.31** |
| | σ | ±0.38 | ±0.47 | ±0.13 | ±0.78 | ±1.12 | ±0.25 |
| IoU (%) | Average | 65.35 | 57.58 | 64.09 | 86.58 | 85.05 | **88.85** |
| | σ | ±3.40 | ±2.22 | ±1.33 | ±1.01 | ±0.29 | ±0.38 |
| Training time (h) | Average | 0.92 | **0.64** | 0.74 | 5.80 | 1.89 | 1.87 |
| | σ | ±0.07 | ±0.08 | ±0.12 | ±0.03 | ±0.07 | ±0.03 |
| FPS (f/s) | Average | 4.01 | 5.45 | 7.11 | 17.29 | 40.04 | **40.50** |
| | σ | ±0.09 | ±0.27 | ±0.08 | ±0.35 | ±0.96 | ±1.44 |
| Parameter size (M) | / | 5.08 | 7.78 | **4.08** | 52.25 | 5.60 | 5.60 |

The bold represents the best performance in the same group.

### 3.2. Model Testing and Results

On the BSL test set, the accuracies of the FPN-M, UNet-M, LinkNet-M, Deeplabv3+ and Deeplabv3+-M-CBAM are shown in Table 5. Figure 13 shows the extraction details for nine example images for BSL-PG (such as Figure 13a–c), BSL-PT (such as Figure 13d,e), buildings (such as Figure 13b–e) and dust-proof net covered areas (such as Figure 13f,g).

**Table 5.** Comparisons for BSL extraction accuracies for different models on BSL test set.

| Model | F1 (%) | Precision (%) | Recall (%) | IoU (%) |
|---|---|---|---|---|
| FPN-M | 60.97 | 68.77 | 70.18 | 53.74 |
| UNet-M | 77.36 | 79.83 | 79.43 | 57.95 |
| LinkNet-M | 79.10 | 81.95 | 80.39 | 69.59 |
| Deeplabv3+ | 79.90 | 79.85 | 88.49 | 75.34 |
| Deeplabv3+-M-CBAM | **88.42** | **87.18** | **92.03** | **85.13** |

The bold represents the best performance in the same group.

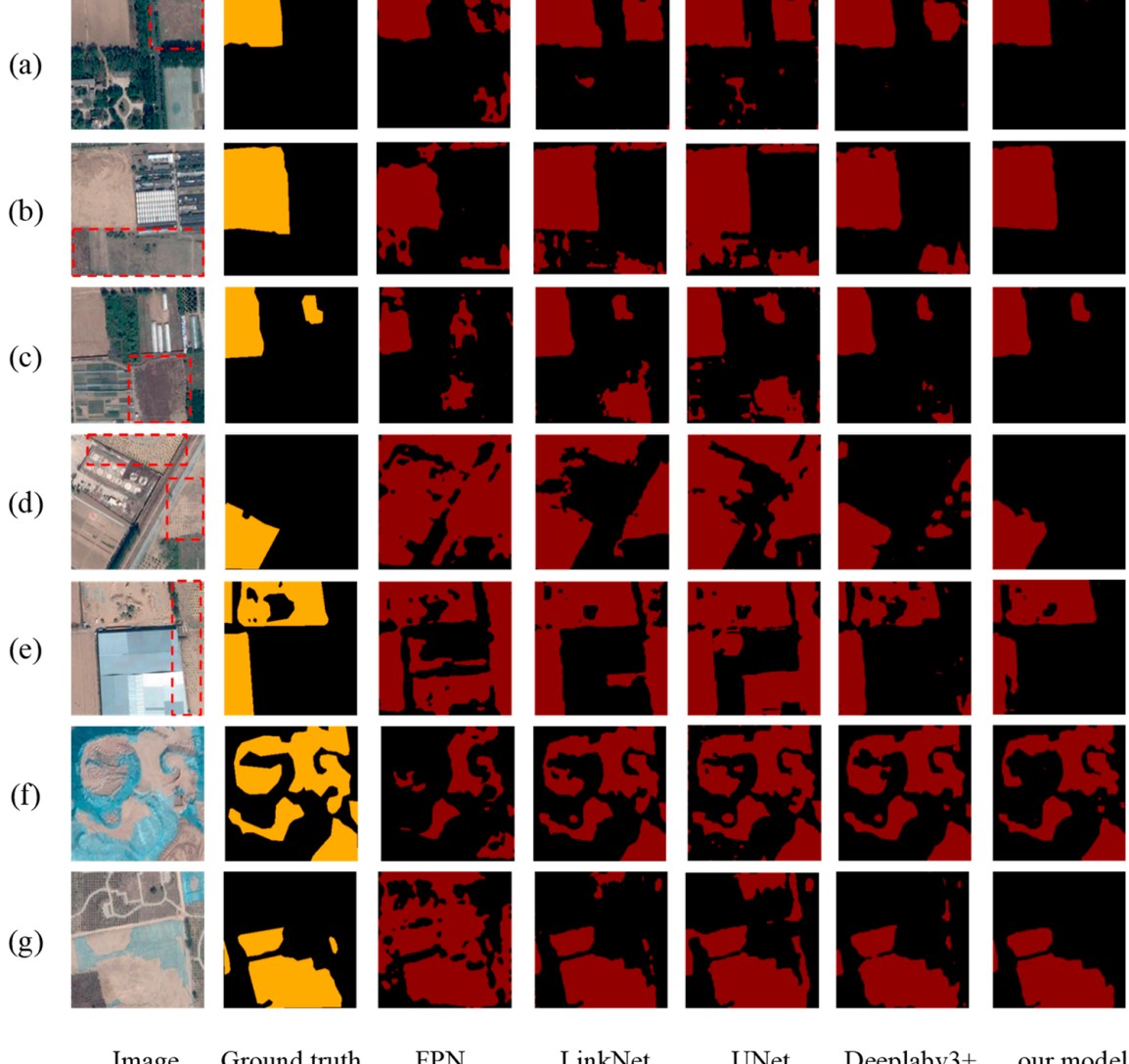

Image  Ground truth  FPN  LinkNet  UNet  Deeplabv3+  our model

**Figure 13.** BSL extraction on the BSL test set for the different models: (**a**–**c**) Interference of BSL-PG; (**d**,**e**) Interference of BSL-PT; (**f**,**g**) Interference of dust-proof net covered areas.

For FPN-M, UNet-M, LinkNet-M and Deeplabv3+, there were false detections in different degrees for the BSL-PG, labeled by the red, dotted boxes in the images, as shown

in Figure 13a–c. Conversely, Deeplabv3+-M-CBAM could distinguish almost all BSL-PG from the background. In Figure 13d, two BSL-PT, as shown by the red, dotted boxes, were mistakenly detected as BSL by FPN-M, UNet-M and LinkNet-M, and buildings were detected as BSL as well. Deeplabv3+ only mistakenly detected one of them as BSL, and Deeplabv3+-M-CBAM had no incorrect detections. In Figure 13e, the BSL-PT in the red, dotted box was extracted by Deeplabv3+, while it was not by Deeplabv3+-M-CBAM. In Figure 13f, all models achieved good extraction. However, in Figure 13g, due to the thinner dust-proof net than that in Figure 13f, the distinction between it and BSL was lower. As a result, there were many false detections in FPN-M and UNet-M, and a small number of false detections in LinkNet-M and Deeplabv3+. Although Deeplabv3+-M-CBAM did not make a mistake, it failed to detect the small area of BSL exposed by the damage of the dust-proof net.

Based on the above analysis, FPN-M, UNet-M and LinkNet-M, as classical semantic segmentation networks, achieved good segmentation results in the case of simple backgrounds and high differentiation from BSL, such as in Figure 13f. However, they had difficulty in distinguishing buildings, BSL-PT and BSL-PG from BSL in complex backgrounds, such as in Figure 13b–d. Since the atrous convolution in Deeplabv3+ can enlarge the receptive field, Deeplabv3+ could distinguish buildings, BSL-PT and BSL-PG from BSL to some extent, but it still cannot meet the segmentation requirements. The Deeplabv3+-M-CBAM greatly improved the ability to distinguish between buildings, BSL-PT and BSL-PG. As can be seen from Table 5, the F1 of Deeplabv3+-M-CBAM was higher than those of the previous models. The test result for F1 was 88.42%, and the FPS was 42.99 f/s, with F1 increased by 27.45%, 11.06%, 9.32% and 8.52% compared to FPN-M, UNet-M, LinkNet-M and Deeplabv3+, respectively.

### 3.3. Automatic Mapping of Large-Scale Transfer Area

In recent years, the application of deep learning technology to automatic mapping for large-scale thematic maps has gradually been applied. Ma [34] used a deep learning method to make a thematic map of greenhouses in China based on high-resolution remote sensing images. Jiang [35] used a deep learning method to quickly create a large map for flood-affected areas. Li [36] used deep learning methods to produce forest canopy maps of combined UAV images with photogrammetric point cloud data.

To further verify the robustness of the Deeplabv3+-M-CBAM model, this study used the transfer area to conduct large-scale automated mapping. As shown in Table 6, taking the result of the visual interpretation as the ground truth, the F1, recall, precision and IoU of our model for BSL was 86.07%, 87.00%, 95.80% and 87.88%, respectively. The time cost of the visual interpretation was 47 min, while the time cost of the automated mapping by our model was only 4 min and 5 s. The efficiency of the mapping of BSL improved by 11.5 times. Figure 14 is a visualization of the extraction results of the transfer area. The transfer test results show that the Deeplabv3+-M-CBAM model had good robustness on data from different data sources and different seasons. In addition, the postprocessing can further improve the accuracy of semantic segmentation results to meet the quality standard of mapping, such as removing pixel noise (filtering or morphological processing) and artificially correcting errors.

**Table 6.** BSL extraction accuracy of Deeplabv3+-M-CBAM in the transfer area.

| Imaging Time | Visual Interpretation Time | Model Extraction Time | F1 (%) | Recall (%) | Precision (%) | IoU (%) |
|---|---|---|---|---|---|---|
| 30 Aug 2021 | 47 min | 4 min 5 s | 86.07 | 87.00 | 95.80 | 87.88 |

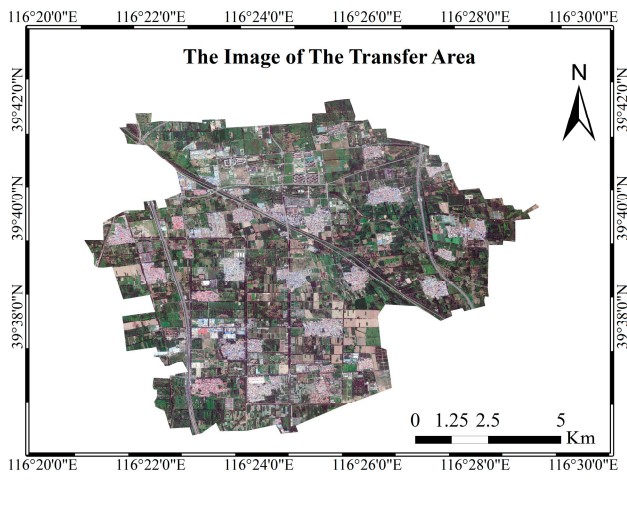

**(a)**

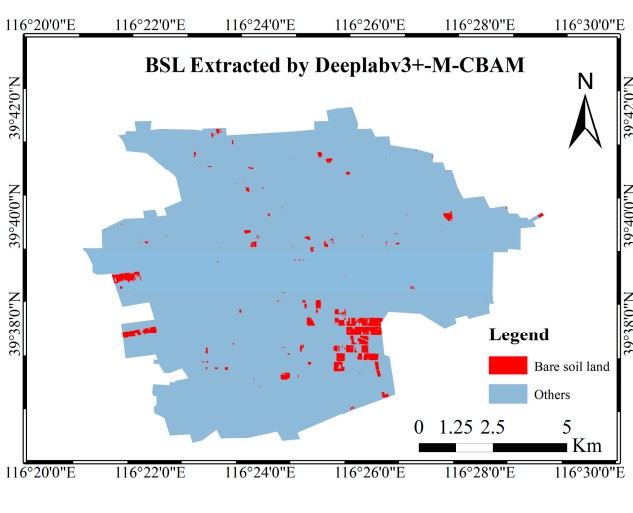

**(b)**                                                                                    **(c)**

**Figure 14.** The results of the large-scale mapping of BSL in the transfer area by visual interpretation and Deeplabv3+-M-CBAM: (**a**) transfer area; (**b**) ground truth of BSL for the transfer area; (**c**) BSL extraction results using our model. The coordinate system is GCS_WGS_1984.

## 4. Discussion

### 4.1. Visualizing Convolutional Networks

To better explain how the deep learning model processes images, this paper visualized the features of different network layers in Figure 15. The redder the pixel in the features, the higher the probability that the pixel belongs to BSL. In shallow features, the boundary details of BSL are rich, but the category information is poor. Although the backbone features have strong category information, the boundary information loss is obvious. After the extraction of the ASPP layers with different rates, the deep features had stronger category information, and the boundary resolution was further reduced. After fusing features in the decoder, the fusion features had obvious category information and clearer boundary information.

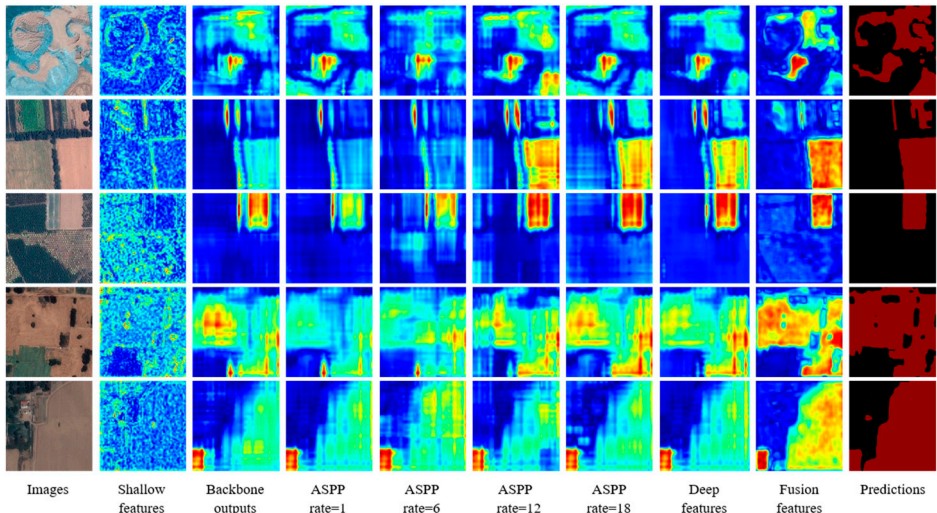

**Figure 15.** Visualizing convolutional networks.

### 4.2. Lighter Semantic Segmentation Mode

Generally, larger receptive fields, deeper networks and more parameters mean that the model has a stronger feature expression ability. On the contrary, the shallower the network and the fewer the number of parameters, the weaker the feature expression ability of the model. However, if the network is too deep and the number of parameters is too large, it will not only increase the detection time and the burden of computer memory but also have the risk of overfitting. Therefore, the model is expected to have as few parameters as possible, be as high precision as possible and have good transferability.

The ASPP module of the Depplabv3+ model can combine multiscale features to expand the receptive field. However, Depplabv3+ could not distinguish BSL-PT and BSL-PG from BSL. The DSC and inverted residual structure of MobileNetV2 can reduce the number of parameters and efficiently integrate features. The CBAM can improve the model's attention to BSL in the process of feature fusion without increasing the number of additional parameters. The extraction results of Deeplabv3+-M-CBAM in the BSL test set show that the CBAM could improve the model's attention to BSL and greatly enhance the distinction ability between buildings, BSL-PG and BSL-PT. Finally, the Deeplabv3+-M-CBAM network achieved a higher model accuracy, with only 10% of the model parameters of Deeplabv3+. A lighter real-time semantic segmentation model can provide effective technical support for monitoring BSL.

### 4.3. Comparison with Other Public Datasets

In order to verify the mapping results of BSL by Deeplabv3+-M-CBAM, the test area was chosen for comparisons with public global land cover products and ground truth by visual interpretation for the same year. Considering the timeliness and accuracy of the data, ESRI10 (2020) and ESA10 (2020) with a 10 m resolution, downloaded from Google Earth Engine, were used.

ESRI10 (2020) was produced by Environmental Systems Research Institute, Inc., (Esri) based on 10 m resolution Sentinel images from 2020 and a deep learning method, which has 10 categories, and the overall accuracy of this product is 85%. ESA10 (2020) is jointly produced based on Sentinel-1 and Sentinel-2 data from 2020 by the European Space Agency (ESA) and some scientific research organizations, which has 10 categories, and its overall accuracy is 74.4%. The classification systems for ESRI10 and ESA10 are shown in Table 7. Bare ground or barren land, including sand, gravel, rock and BSL, is classified in these two classification systems.

**Table 7.** The classification system of ESRI10 and ESA10.

|  | ESRI10 | ESA10 |
|---|---|---|
|  | Water | Water bodies |
|  | Trees | Forest |
|  | Grass | Grasslands |
|  | Flooded vegetation | Wetlands |
| Categories | Crops | Cropland |
|  | Scrub/shrub | Shrublands |
|  | Built area | Impervious |
|  | Bare ground | Barren land |
|  | Snow/ice | Snow and ice |
|  | Clouds | Tundra |

The comparisons of the mapping results using our model, ESRI10 (2020) and ESA10 (2020) with ground truth data are shown in Figure 16. It was found that there were also great differences among the different mapping results. For ESRI10 (2020), the classification results were coarse, and there were only a few bare land patches. For ESA10 (2020), there were many more bare land patches than that of the results using Deeplabv3+-M-CBAM, because there were only BSL patches extracted by Deeplabv3+-M-CBAM.

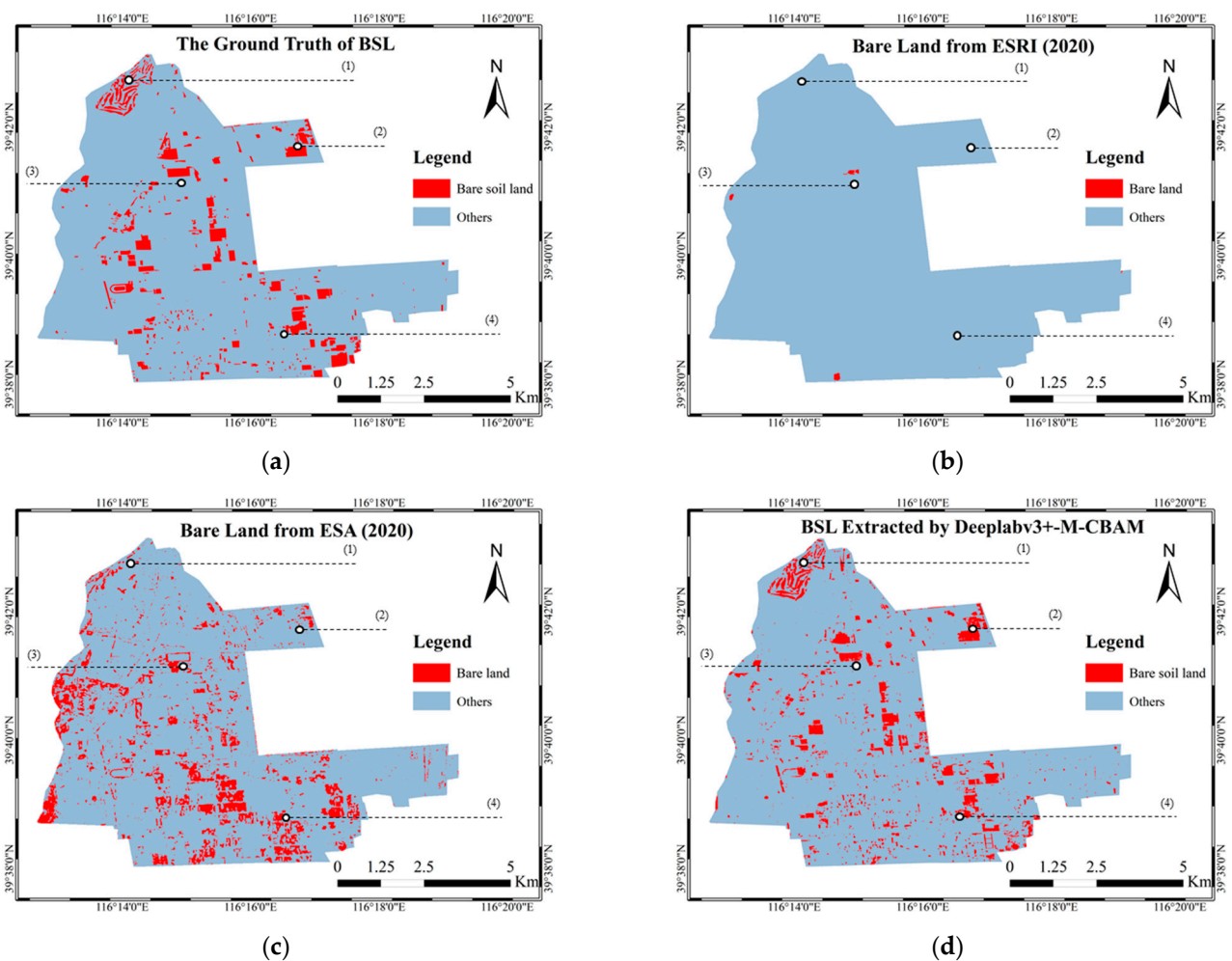

**Figure 16.** BSL-related mapping results for the different LULC products: The coordinate system of the maps is GCS_WGS_1984.

We selected four typical regions (corresponding to four images) from the test area for more detailed comparisons. The results are shown in Figure 17. For ESRI10, only one

region of bare land in the image shown in Figure 17(3) was classified. For ESA10, the boundary fineness of bare land was insufficient, and there were obvious omissions for large areas of BSL in the images shown in Figure 17(1),(2). Some areas of greenhouses and buildings were falsely detected as bare land in the images in Figure 17(3),(4). Some of the omissions were probably due to the different acquisition times of the images. Yet, the results of Deeplabv3+-M-CBAM were the closest to the ground truth, except for some fragmental patches of BSL and noise.

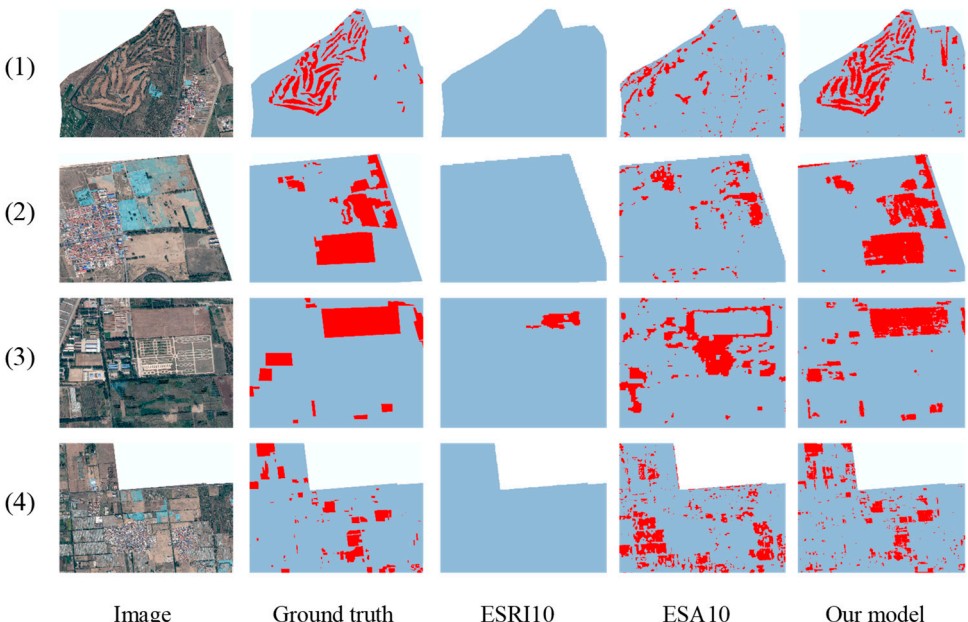

**Figure 17.** Different results for bare land for the different classifications: (1–4) Four typical regions.

From the comparisons, it can be found that there were problems of insufficient accuracy and coarse-grained definition for bare land, although the current public datasets contain bare land. For the results of the BSL extraction by Deeplabv3+-M-CBAM from high-resolution remote sensing images, both the boundary accuracy and class accuracy were the closest to the ground truth.

## 5. Conclusions

The precise detection of BSL is of great significance to improve the utilization rate of land resources and ecological environmental governance. Current land cover products are mostly produced based on medium- or low-resolution images, such as MODIS, Landsat and Sentinel, and the granularity and accuracy of the classes cannot meet the requirements of regional BSL extraction and monitoring. High-resolution images can provide more detailed information on objects, but its limited number of spectral bands results in some difficulties in separating BSL and buildings with similar spectral characteristics. Moreover, complex backgrounds with rich semantic information pose challenges to traditional methods. In the process of the fine management of BSL, BSL-PT and BSL-PG are formed due to the planting of grass and trees, which makes the background more complicated and interferes with the extraction of BSL.

In summary, this study proposes a lighter semantic segmentation model combined with the CBAM. The improved Deeplabv3+ model (Deeplabv3+-M-CBAM) extracts BSL from complex backgrounds and performs well in test accuracy. Compared with mainstream models, Deeplabv3+-M-CBAM had the highest ability to distinguish BSL from BSL-PG, BSL-PT and buildings. Due to the 90% reduction in the parameters, Deeplabv3+-M-CBAM can be deployed to run on machines with more limited resources. This study provides technical support for the governance of BSL by artificial intelligence technology. Meanwhile,

it will enrich the classification granularity of traditional LULC classification and promote the fine classification of LULC in the near future.

Certainly, there are still two further studies on the datasets and postprocessing for the results that can be considered. In regard to the data, the cloud coverage of the image data we used was 0%. However, in most cases, the acquired images often have cloud coverage. Therefore, in order to enhance the robustness of the model, cloud samples can be added into the BSL dataset for the training of the model. In regard to the postprocessing of the results of the model, since the BSL extraction results were pixel-based raster data, it should be processed according to the different minimum statistic units, for example, $3 \times 3$ pixels, to remove the pixels smaller than this size and to satisfy the different demands for BSL governance. These improvements will provide better technical support for perennial BSL monitoring and more detailed demands.

**Author Contributions:** Conceptualization, C.H., Y.L. and Y.R.; methodology, C.H. and Y.L.; software, C.H.; validation, Y.R. and Y.L.; formal analysis, Y.L.; investigation, Y.R., S.L., L.Y. and Y.L.; resources, Y.L.; data collection and processing, Y.R. and S.L.; writing—original draft preparation, C.H. and Y.L.; writing—review and editing, Y.R. and C.H; visualization, D.W.; supervision, Y.L, Y.R. and S.L.; project administration, D.W. and Y.L. All authors have read and agreed to the published version of the manuscript.

**Funding:** This research was funded by the Project of Dynamic Remote Sensing Monitoring of Bare Soil in Daxing District, Beijing, China (grant number: DXCG_21_0904).

**Data Availability Statement:** Not applicable.

**Acknowledgments:** The authors are very grateful to the people who helped in the acquisition of the satellite images for this article and are also grateful to the anonymous reviewers for their helpful comments and suggestions.

**Conflicts of Interest:** We declare that we have no conflict of interest.

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
