# Peer review of "Automatic Extraction of Bare Soil Land from High-Resolution Remote Sensing Images Based on Semantic Segmentation with Deep Learning"

_remotesensing, doi:10.3390/rs15061646_

Round 1

Reviewer 1 Report

The paper develops a new CBAM-based architecture for monitoring for bare soil land based on semantic segmentation of high-resolution aerial images using deep learning. A new dataset is also collected to test the proposed architecture. Experimental results are performed to demonstrate that the architecture is effective and leads to improved performance.
  The paper is addressing a practically important challenge and reads well. I have the following comments to be addressed for the next round of reviews.     1. Please expand section 2.2.3 and provide more substantive explanations about transfer learning.   2. Please repeat the experiments several times and report both the average performance and the standard deviation to make the comparison statistically meaningful.   3. Please report Intersection over Union in your experiments as it is a common metric for evaluation of the quality of segmentation.   4. Semantic segmentation is known to require a significant amount of annotated data which limits using UNet. A group of algorithms addresses this challenge using domain adaptation which can be considered a type of transfer learning algorithm:      a. Guizilini, V., Li, J., AmbruÈ™, R. and Gaidon, A., 2021. Geometric unsupervised domain adaptation for semantic segmentation. In Proceedings of the IEEE/CVF International Conference on Computer Vision (pp. 8537-8547).      b. Stan, S. and Rostami, M., 2021, May. Unsupervised model adaptation for continual semantic segmentation. In Proceedings of the AAAI conference on artificial intelligence (Vol. 35, No. 3, pp. 2593-2601).      c. Toldo, M., Michieli, U. and Zanuttigh, P., 2021. Unsupervised domain adaptation in semantic segmentation via orthogonal and clustered embeddings. In Proceedings of the IEEE/CVF Winter Conference on Applications of Computer Vision (pp. 1358-1368).      d. Marsden, R.A., Wiewel, F., Döbler, M., Yang, Y. and Yang, B., 2022, July. Continual unsupervised domain adaptation for semantic segmentation using a class-specific transfer. In 2022 International Joint Conference on Neural Networks (IJCNN) (pp. 1-8). IEEE.      The above work should be discussed in the introduction section to provide the reader with background about addressing the practical challenges of data annotation in semantic segmentation.   5. Please release the dataset on a public domain so other researchers who want to work on this important application.   6. Please release the code for this work into the public domain so other researchers are able to reproduce the results.

Author Response

Response to Reviewer’s Comments 1

Comments and Suggestions for Authors

The paper develops a new CBAM-based architecture for monitoring for bare soil land based on semantic segmentation of high-resolution aerial images using deep learning. A new dataset is also collected to test the proposed architecture. Experimental results are performed to demonstrate that the architecture is effective and leads to improved performance.
The paper is addressing a practically important challenge and reads well.

Response: Thank you very much for the positive comments and constructive suggestions. We have tried our best to revise the manuscript according to your kind and construction comments and suggestions. We sincerely hope that this revised manuscript has addressed all your comments and suggestions.

Point 1: Please expand section 2.2.3 and provide more substantive explanations about transfer learning.

Response1: Thanks for your suggestion. We have expanded section 2.2.3 according to your suggestion. To provide more information about model-based transfer learning, we have made a new figure (Figure 9) as shown as below. “Transfer learning is a method that aims to transfer knowledge from a source domain to improve a model’s performance or minimize the number of labeled examples required in a target domain. Model-based transfer learning is a kind of transfer learning solution. It is a way to continue learning based on the previous learning for the model. For deep learning, this refers to the fact that a model is firstly trained on an unrelated dataset of task A, and uses the training result as the pretrained model for task B to initialize the model. …An effective deep learning model requires a large amount of annotated data. However, compared to large datasets, such as VOC12, the BSL dataset has a small number of labeled samples.…”

Figure 9. Model-based transfer learning

Point 2: Please repeat the experiments several times and report both the average performance and the standard deviation to make the comparison statistically meaningful.

Response2: Thanks for your constructive suggestion. We have repeated the experiments five times and reported both the average performance and the standard deviation in section 3.1.

Table4. The performance comparison for training the different models on BSL validation set.

Indexes

/

FPN-M

UNet-M

LinkNet-M

Deeplabv3+

Deeplabv3+-M

Deeplabv3+-M-CBAM

F1(%)

Average

69.44

61.95

68.32

91.96

91.09

93.49

σ

±3.35

±2.27

±1.26

±0.76

±0.97

±0.24

Precision(%)

Average

70.71

63.39

69.67

92.53

92.49

93.89

σ

±3.58

±1.91

±1.33

±0.48

±0.34

±0.40

Recall(%)

Average

93.58

92.97

93.17

92.13

91.64

94.31

σ

±0.38

±0.47

±0.13

±0.78

±1.12

±0.25

IoU(%)

Average

65.35

57.58

64.09

86.58

85.05

88.85

σ

±3.40

±2.22

±1.33

±1.01

±0.29

±0.38

Training time (h)

Average

0.92

0.64

0.74

5.80

1.89

1.87

σ

±0.07

±0.08

±0.12

±0.03

±0.07

±0.03

FPS(f/s)

Average

4.01

5.45

7.11

17.29

40.04

40.50

σ

±0.09

±0.27

±0.08

±0.35

±0.96

±1.44

Parameter size(M)

/

5.08

7.78

4.08

52.25

5.60

5.60

Point 3: Please report Intersection over Union in your experiments as it is a common metric for evaluation of the quality of segmentation.

Response3: Thanks for your suggestion. We have added the Intersection over Union (IoU) of 6 models in Figure 12 and Table 4.

(a)

(b)

Figure12. Accuracy curves of the 6 models: (a) F1 curves on the validation set; (b) IoU curves on the validation set.

Point 4: Semantic segmentation is known to require a significant amount of annotated data which limits using UNet. A group of algorithms addresses this challenge using domain adaptation which can be considered a type of transfer learning algorithm:     

  1. Guizilini, V., Li, J., AmbruÈ™, R. and Gaidon, A., 2021. Geometric unsupervised domain adaptation for semantic segmentation. In Proceedings of the IEEE/CVF International Conference on Computer Vision (pp. 8537-8547).
  2. Stan, S. and Rostami, M., 2021, May. Unsupervised model adaptation for continual semantic segmentation. In Proceedings of the AAAI conference on artificial intelligence (Vol. 35, No. 3, pp. 2593-2601).
  3. Toldo, M., Michieli, U. and Zanuttigh, P., 2021. Unsupervised domain adaptation in semantic segmentation via orthogonal and clustered embeddings. In Proceedings of the IEEE/CVF Winter Conference on Applications of Computer Vision (pp. 1358-1368).
  4. Marsden, R.A., Wiewel, F., Döbler, M., Yang, Y. and Yang, B., 2022, July. Continual unsupervised domain adaptation for semantic segmentation using a class-specific transfer. In 2022 International Joint Conference on Neural Networks (IJCNN) (pp. 1-8). IEEE.

The above work should be discussed in the introduction section to provide the reader with background about addressing the practical challenges of data annotation in semantic segmentation.

Response4: Thanks for your constructive suggestion. We have added above work in the introduction section. “…However, semantic segmentation requires a significant amount of annotated data , which limits the use of deep learning models. To addresses this challenge, transfer learning is proposed. Transfer learning can be divided into instance-based transfer learning, feature-based transfer learning, model-based transfer learning and relation-based transfer learning [14]. Domain adaptation is another term commonly used in transfer learning, and many studies are addressing this challenge of limited annotated data [15–18]. For cases where the datasets types of the source and target domains are homogeneous (for example, photos of roads in different countries), domain adaptation can transfer their domain invariant features. If the data types of the source and target domains are heterogeneous (for example, photos taken by the a phone and remote sensing images), model-based transfer learning is more feasible…”

Point 5: Please release the dataset on a public domain so other researchers who want to work on this important application.

Response5: Thanks for your suggestion. The release of the dataset will be considered after the patent application and article for the relevant technology is approved. Please feel free to contact us by email if necessary. It can be given after evaluation at this stage.

Point 6: Please release the code for this work into the public domain so other researchers are able to reproduce the results.

Response6: Thanks for your suggestion. When the patent application and article for the relevant technology is approved, the authors will choose to make the code public and will publish the trained models together.

Reviewer 2 Report

1.the innovation of the method should be clearly rewriten, and give the details of reasons why combinations of different modules.

2. all of the maps doesn't meet cartographic standards.

3.the conclusion should be rewriten more concise.

4. some reference are missing.

Author Response

Response to Reviewer’s Comments 2

Point 1: the innovation of the method should be clearly rewritten, and give the details of reasons why combinations of different modules.

Response1: Thanks for your constructive suggestion. We have added two subheadings in section 2.2.2 and given more details of reasons why combinations of different modules.

“(1) A lighter backbone network—MobileNetV2

(2) An optimized backbone network—M-CBAM”

“…decrease the number of operations and memory needed of Deeplabv3+, we replaced its Xception backbone network with a more lightweight network named MobileNetV2…”  “…Semantic segmentation focuses on both category information (“what”) and boundary information (“where”). In addition, the channel and spatial attention modules of the CBAM can learn “what” and “where” to attend in the channel and spatial axes, respectively.…In the M-CBAM structure, the category information and boundary information of BSL are enhanced twice by the CBAM.”

“…In summary, MobileNetV2 can decrease the number of operations and memory needed by losing a small amount of precision. The CBAM can focus on BSL and suppress background information. Based on the above points, the Deeplabv3+-M-CBAM was constructed.”

Point 2: all of the maps doesn't meet cartographic standards.

Response2: Thank you for pointing this out. We have carefully checked all the maps and have determined their map border, title, mathematical element (scale, coordinate system), compass, legend. And the coordinate system of maps is described in their figure legend. The revised maps have replaced the original maps according to your suggestion.

Point 3: the conclusion should be rewritten more concise.

Response3: Thanks for your suggestion. We have simplified the conclusion according to your suggestion.

Point 4: some reference are missing.

Response4: Thank you for pointing this out. We have added the references where the references are missing, and then carefully checked the other reference as well.

Reviewer 3 Report

This study proposes an automatic extraction of bare soil land from high-resolution remote sensing images based on semantic segmentation with deep learning. Its main contribution consists in the improvement of the network structure of the original Deeplabv3+ model turning it more suitable for BSL extraction from high-resolution remote sensing images.

The document is sometimes not easy to read and follow.

The English needs major grammar and spell checking.

The document is well supported with references although many are old.

The subject of the paper has great potential of application.

Any acronym presented in the document should be described before it is used.

In line 67 please correct “…Deep learning method has been widely used in natural images recent years…”

In line 70 authors should not use “etc.”. Please correct “…end-to-end for learning and etc…”

In line 71 please correct “… They are many …”

Authors should describe in more detail the dataset images labeling process including the tools used.

In subsection 2.2 authors state that replaced the Xception backbone network with a more lightweight network but in the Abstract was said it was replaced due to overfitting. Please standardize.

In line 295 did you mean “Frames Per Second (FPS)”?! Please verify.

Please correct line 317 to English text.

Authors should include in the description of the used algorithms the language in which they were programmed.

In subsection “4.2 Comparison with other public datasets” authors should include a performance table like Table 6.

Author Response

Response to Reviewer’s Comments 3

Comments and Suggestions for Authors

This study proposes an automatic extraction of bare soil land from high-resolution remote sensing images based on semantic segmentation with deep learning. Its main contribution consists in the improvement of the network structure of the original Deeplabv3+ model turning it more suitable for BSL extraction from high-resolution remote sensing images.

The subject of the paper has great potential of application.

Response: Thank you very much for the positive comments and constructive suggestions. We have tried our best to revise the manuscript according to your kind and construction comments and suggestions. We sincerely hope that this revised manuscript has addressed all your comments and suggestions.

Point 1: The document is sometimes not easy to read and follow. The English needs major grammar and spell checking.

Any acronym presented in the document should be described before it is used.

In line 67 please correct “…Deep learning method has been widely used in natural images recent years…”

In line 70 authors should not use “etc.”. Please correct “…end-to-end for learning and etc…”

In line 71 please correct “… They are many …”

In subsection 2.2 authors state that replaced the Xception backbone network with a more lightweight network but in the Abstract was said it was replaced due to overfitting. Please standardize.

In line 295 did you mean “Frames Per Second (FPS)”?! Please verify.

Please correct line 317 to English text.

Response1: Thanks for your constructive suggestion. We have modified these imperfect expressions according to your suggestion. In addition, to ensure the correct language, our manuscript has undergone extensive English revisions.

Point 2: The document is well supported with references although many are old.

Response2: Thanks for your suggestion. We have replaced the old references as much as possible. We have added some new references and still retained a few classic references.

Point 3: Authors should describe in more detail the dataset images labelling process including the tools used.

Response3: Thanks for your suggestion. We have expanded the detail in section 2.1.2. “…The semantic annotation was conducted in EISeg (Efficient Interactive Segmentation), which is an efficient and intelligent interactive segmentation annotation software developed based on PaddlePaddle.…”

Point 4: Authors should include in the description of the used algorithms the language in which they were programmed.

Response4: Thank you for pointing this out. We have added description of the used algorithms the language in section 2.3. “…The software configuration was Win10, Python 3.6 and torch1.8.0+cu111. …”

Point 5: In subsection “4.2 Comparison with other public datasets” authors should include a performance table like Table 6.

Response5: Thanks for your suggestion. The two public LULC products’ Bare Land includes BSL, gravel and rock. Thus, this comparison is to discuss problems of insufficient accuracy and coarse-grained definition for bare land although the current public datasets contain bare land. BSL belongs to the Bare Land, so it isn’t appropriate to evaluate them together by one indicator.

Reviewer 4 Report

This article uses DeeplabV3+ as the main body, combined with the MobileNetV2 backbone network, to perform bare soil land segmentation, which improves both accuracy and speed.

1. In lines 78 and 79, you mentioned "It allows the input image on arbitrary scales to construct multi-receptive fields.", I can't understand the way this sentence is done, please revise this sentence carefully, it should be correct Arbitrary scales performed by feature maps.

2. In lines 80 and 81, you mentioned: "solve the problem of multi-scale segmentation". I think your description of "multi-scale segmentation" is inaccurate.

3. In your Figure 5, the rates of ASPP are 6, 12, and 18 respectively, but your cut image size is 256×256, so I think this rate may not be appropriate, and you need to explain in detail.

4. In Section 2.2.3, the CBAM module was inserted into the MobileNetV2 model according to your previous article. So I think this part of your description is inaccurate, please modify it.

Author Response

Response to Reviewer’s Comments 4

Comments and Suggestions for Authors

This article uses DeeplabV3+ as the main body, combined with the MobileNetV2 backbone network, to perform bare soil land segmentation, which improves both accuracy and speed.

Response: Thank you very much for the positive comments and constructive suggestions. We have tried our best to revise the manuscript according to your kind and construction comments and suggestions. We sincerely hope that this revised manuscript has addressed all your comments and suggestions.

Point 1: In lines 78 and 79, you mentioned "It allows the input image on arbitrary scales to construct multi-receptive fields.", I can't understand the way this sentence is done, please revise this sentence carefully, it should be correct Arbitrary scales performed by feature maps.

Response1: Thanks for your suggestion. We have modified it to “It allows for the input image at arbitrary scales to be performed by feature maps.”

Point 2: In lines 80 and 81, you mentioned: "solve the problem of multi-scale segmentation". I think your description of "multi-scale segmentation" is inaccurate.

Response2: Thank you for pointing this out. We have modified it to “…several atrous convolution modules with different expansion rates are used to capture the multi-scale context. ”.

Point 3: In your Figure 5, the rates of ASPP are 6, 12, and 18 respectively, but your cut image size is 256×256, so I think this rate may not be appropriate, and you need to explain in detail.

Response3: Thanks for your suggestion, which is highly appreciated. To show the details of the different rates of ASPP, we have added a section 4.1 and have visualized the feature maps in Figure 15. “To better explain how the deep learning model processes images, this paper visualized the features of different network layers. The redder the pixel in the features, the higher the probability that the pixel belongs to BSL. In shallow features, the boundary details of BSL are rich, but the category information is poor. Although the backbone features have strong category information, the boundary information loss is obvious. After the extraction of the ASPP layers with different rates, the deep features had stronger category information, and the boundary resolution was further reduced. After fusing features in the decoder, the fusion features had obvious category information and clearer boundary information.” It can be seen from the details of extraction process that the BSL information is well extracted. Therefore, these rates are suitable for image size of 256×256 from the extraction process and the prediction.

Figure 15. Visualizing convolutional networks

Point 4: In Section 2.2.3, the CBAM module was inserted into the MobileNetV2 model according to your previous article. So I think this part of your description is inaccurate, please modify it.

Response4: Thanks for your suggestion. In Figure 8 of section 2.2.2, we introduce that CBAM respectively processed the input and output features of MobileNetV2. In this progress CBAM doesn’t destroy the internal structure of MobileNetV2. Thus, in section 2.2.3, the pretrained model of MobileNetV2 can be used for retraining.

To avoid ambiguity, we changed “transplant…into…” to “merged…to…” as “In this study, we merged the CBAM and MobileNetV2 to construct MobileNetV2–CBAM (M-CBAM).”

Figure 8. The optimization of MobileNetV2 network.

Round 2

Reviewer 1 Report

The authors have addressed my primary concerns. My only recommendation is that please bear in mind to release the code and the data when the patent application materialized. 

Reviewer 2 Report

1. P15:445-446: As shown in Table 6, taking the result of the visual interpretation as the ground truth, the F1, recall, precision and IoU  of our model for BSL was 86.07%, 87.00%, 95.80% and 87.88%, respectively. You shoulld give the postprocessing methods to meet the quality standard of final result which refers to recall (-13%), precision(-4.2%) and IoU(-12.12%).

2.The sentences of the paper should be carefully polished again, not only rectifying  the grammar, but also the ways of expression.

Reviewer 4 Report

You have revised the question I mentioned and also explained my doubts.
